# ReWiTe: Realistic Wide-angle and Telephoto Dual Camera Fusion Dataset via Beam Splitter Camera Rig

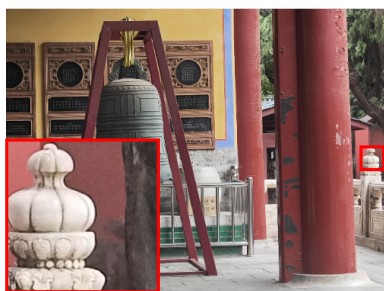 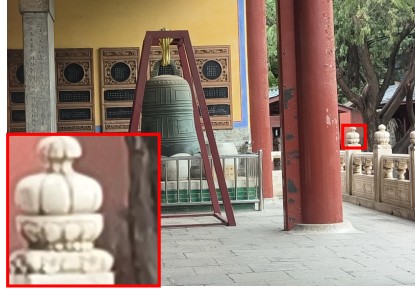 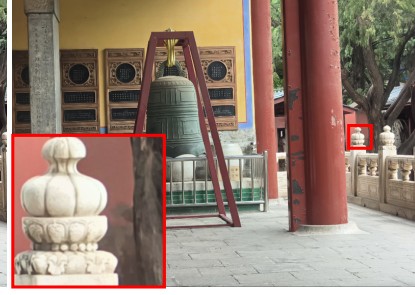

| Telephoto | Wide-angle | Ground-truth |

Figure 1: Examples of the input pair of telephoto (T) and wide-angle (W) images from the dual cameras, along with the ground-truth (GT) image in the proposed ReWiTe dataset. Notably, all three images are captured by real cameras and none of them are synthesized. The quality of the GT image is as high as that of the input T image. In addition, the GT image shares the same optical path and field of view (FOV) with the input W image, providing annotations for the input W image at every pixel. The red boxes indicate the enlarged image regions.

## ABSTRACT

The fusion of images from dual camera systems featuring a wide-angle and a telephoto camera has become a hotspot problem recently. By integrating simultaneously captured wide-angle and telephoto images from these systems, the resulting fused image achieves a wide field of view (FOV) coupled with high-definition quality. Existing approaches are mostly deep learning methods, and predominantly rely on supervised learning, where the training dataset plays a pivotal role. However, current datasets typically adopt a data synthesis approach, where the wide-angle inputs are synthesized rather than captured using real wide-angle cameras, and the ground-truth image is captured by wide-angle cameras whose quality is substantially lower than that of input telephoto images captured by telephoto cameras. To address these limitations, we introduce a novel hardware setup utilizing a beam splitter to simultaneously capture three images, i.e. input pairs and ground-truth images, from two authentic cellphones equipped with wide-angle and telephoto dual cameras. Specifically, the wide-angle and telephoto images captured by cellphone 2 serve as the input pair, while the telephoto image captured by cellphone 1, which is calibrated to match the optical path of the wide-angle image from cellphone 2, serves as the ground-truth image, maintaining quality on par with the input telephoto image. Experiments validate the efficacy of our newly introduced dataset, named ReWiTe, which can

significantly enhance the performance of various existing methods for the real-world wide-angle and telephoto dual image fusion task.

## CCS CONCEPTS

• **Computing methodologies**;

## KEYWORDS

Dataset, Beam Splitter, Realistic, Wide-angle and Telephoto

## 1 INTRODUCTION

The deployment of dual camera systems with a wide-angle (**W**) camera and a telephoto (**T**) camera have become widespread in mainstream smartphones, e.g. iPhone 15, HuaWei Mete 60, OPPO find x7. As Fig. 1 shows, the images from the **W** camera have wide field of view (FOV), and the images from the **T** camera have high definition quality. It is a natural idea to let the dual cameras simultaneously shoot images and fuse these two images to generate the result with both wide FOV and high definition quality. Consequently, this problem has emerged as a recent hotspot in the field [20].

In existing state-of-the-art approaches, some utilize the traditional image blending framework [6]. However, this method is limited because the input **T** image has a smaller field of view (FOV) than the input **W** image. Consequently, it can only enhance the center overlapping regions and cannot enhance the un-overlapping regions. Deep learning based super-resolution methods [14, 20] exhibit strong capabilities and substantial potential in addressing this problem. The mainstream methods among them rely on supervised learning, where the training data plays a crucial role in training a robust model.

The current **W** and **T** dual camera fusion datasets are primarily synthetic, e.g. CameraFusion [20]. As shown in Fig. 2 and Table 1, they use dual camera systems to shoot **W** and **T** images. The shot **T**

*ACM MM, 2024, Melbourne, Australia*

© 2024 Copyright held by the owner/author(s). Publication rights licensed to ACM.
ACM ISBN 978-x-xxxx-xxxx-x/YY/MM
https://doi.org/10.1145/nnnnnnn.nnnnnnn

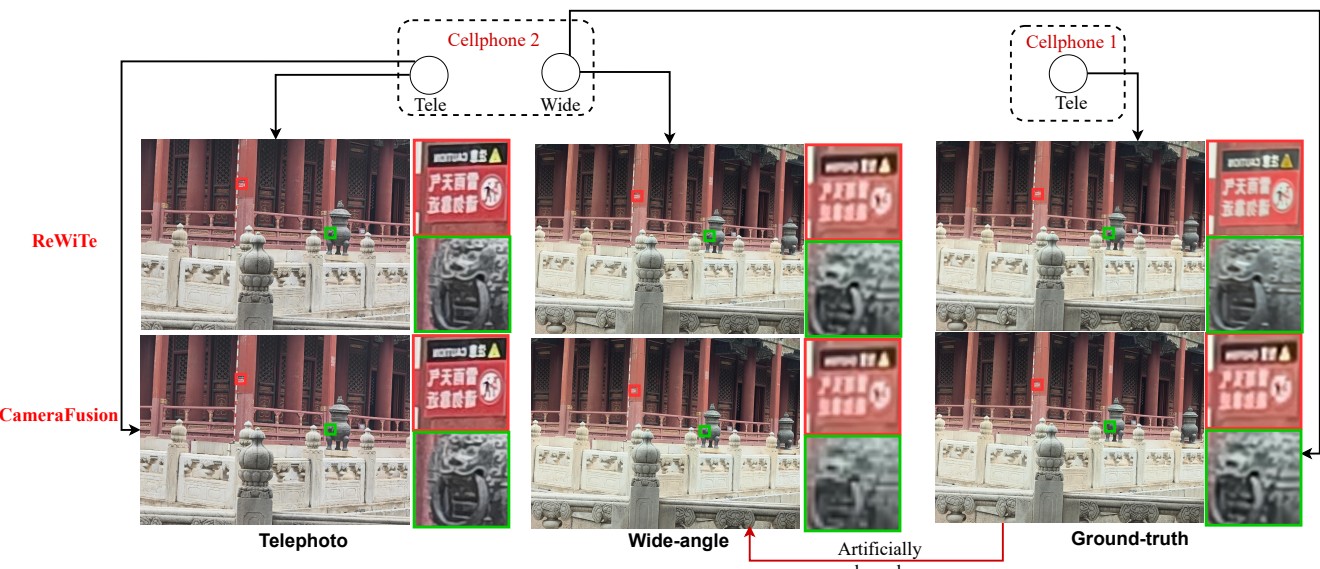

**Figure 2: A comparison between our ReWiTe dataset and the synthesized CameraFusion dataset. In ReWiTe, all input pairs of telephoto (T) and wide-angle (W) images are captured using real cameras, with the ground-truth (GT) image captured by a real T camera as well. In contrast, in CameraFusion, the input W image is synthesized from the GT image through artificial degradation, and the GT image is captured by a W camera, which has significantly lower quality compared to the T camera. 'Wide' is short for the wide-angle camera, and 'Tele' is short for the telephoto camera.**

image serves as the input **T** image, while the shot **W** image serves as the ground-truth (**GT**) image. To synthesize the input **W** image, the **GT** image undergoes distortions such as downsampling and noise addition. However, since the **GT** image is shot by the **W** camera, the quality of input **T** image is significantly higher than that of the **GT** image. This quality difference can confuse the network during training, leading it to produce outputs similar to the **GT** image despite the reference **T** image which has superior quality. Consequently, the trained model may incorrectly utilize the high-quality reference to generate lower-quality results, which is not the intended outcome. In addition, the quality difference between the synthesized input **W** image and the **GT** image is artificially introduced, through operations like downsampling and noise addition. These synthetic distortions can hardly accurately simulate the real quality distortion occurring throughout the complex imaging pipeline, including processes like optical lens effects, demosaicing, tone adjustment, image enhancement, denoising, etc.

As mentioned by [11], due to the inherent characteristics of deep learning networks and the limitations associated with synthetic datasets, the necessity for realistic training data in this domain becomes apparent. This motivates us to construct a realistic dataset in this paper. Our approach involves using a real dual-camera system to capture the input pair of **W** and **T** images, ensuring that their qualities remain realistic without artificial degradation. Additionally, we employ another **T** camera to capture an image with the same optical path as the **W** image, which serves as the **GT** image for the **W** input. This approach guarantees that the **GT** image is realistic and maintains a quality level as high as the reference **T** image. By doing so, we encourage the training of deep models to fully leverage the high-quality details present in the **T** input to

enhance the **W** input. Example images from ReWiTe are shown in Fig. 3.

To build the ReWiTe dataset, we build a hardware, as shown in Fig. 4. Our hardware includes two cellphones, a beamsplitter, and a fixation device. The captured images from the **T** camera of cellphone No.1 is used as the **GT** image. The **W** and **T** cameras of cellphone No. 2 capture images as the input **W** and **T** images. The beamsplitter divides the optical path into a 50%-50% split, allowing the **T** camera of cellphone No. 1 and the **W** camera of cellphone No. 2 to capture images from the same optical path. We conduct the calibration for image alignment and mitigating their differences in color and scale.

In addition to constructing the ReWiTe dataset, we conduct experiments to evaluate the performance of various state-of-the-art (SOTA) methods on ReWiTe. Furthermore, we utilize the training data provided by ReWiTe to retrain these SOTA methods and evaluate their performance once more. Quantitative and qualitative results demonstrate that our ReWiTe dataset can enhance the accuracy of various state-of-the-art algorithms compared to models trained on synthetic datasets such as CameraFusion [20].

Contributions: To the best of our knowledge, this is the first realistic dataset for the telephoto and wide-angle dual camera image fusion task. It provides 342 sets of authentic input telephoto images, input wide-angle images and ground-truth images for training and testing purposes.

## 2 RELATED WORK

### 2.1 Related datasets

In related image enhancement problems for camera imaging, a common method to build datasets is to generate synthetic datasets

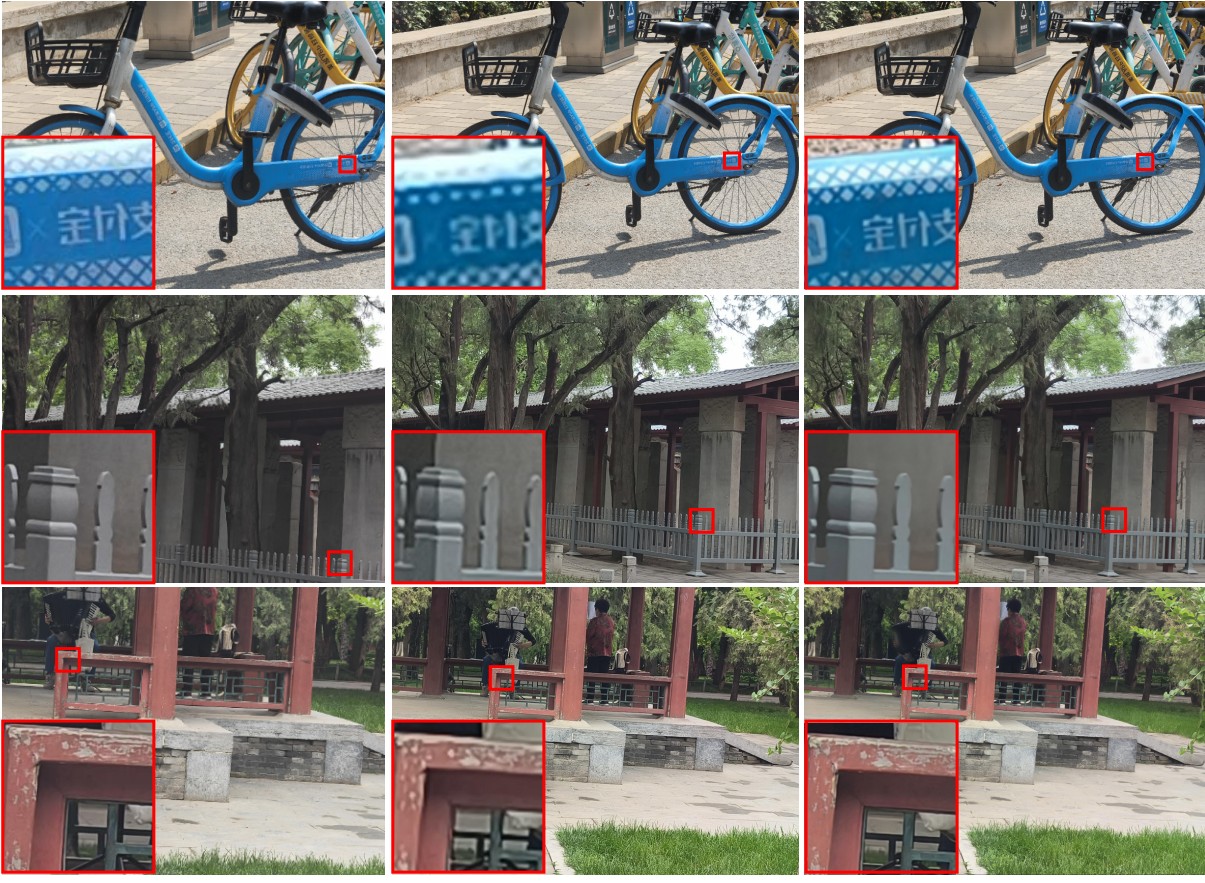

|  Telephoto | Wide-angle | Ground-truth |

**Figure 3: Examples of the input pair of Telephoto (T) and Wide-angel (W) images and the Ground-truth (GT) image of the proposed ReWiTe dataset. The red boxes indicate the enlarged image regions.**

for training and testing, such as CUFED5 [27] for reference-based super-resolution, DIV2K [3], Set5 [4], Set10 [29], Urben100 [8] for single image super resolution, CameraFusion [20] and MiddleBury [17] for **W** and **T** dual camera fusion. In these datasets, the ground truth images are artificially degraded to generate the input images. The degradation process involves operations like downsampling, adding blur, introducing random noise, and applying compression noise. However, the manual degradation process can hardly simulate the imaging differences among cameras with varying imaging pipelines and parameters, leading to a disconnect between training data and real-world application scenarios.

To the best of our knowledge, the problems with accessible real annotated data include 1) obtaining real annotated data for single image denoising through shooting a burst of images and then average [1, 2], and 2) obtaining real annotated data for single-image super-resolution through a beam splitter system [11]. The latter one motivates us to also use a beam splitter to build the ReWiTe dataset.

Table 1 shows the summary of exiting multi-image and single-image super resolution datasets. Among the existing datasets, none of them can provide all the real images of **T**, **W**, and **GT**. This motivates us to establish real annotated data for the emerging problem of **W** and **T** dual camera fusion.

**Table 1: Summary of existing image fusion and super-resolution datasets. While realistic datasets exist for the single-image super-resolution task, e.g. RealSR [5] and ImagePairs [11], there is a lack of realistic datasets for the W and T dual camera fusion task. The proposed ReWiTe dataset aims to address this limitation.**

| Dataset | Telephoto (**T**) | Wide-angle (**W**) | Ground-truth (**GT**) |
|---|---|---|---|
| DIV2K[3] | N/A | Synthetic | Real |
| Urban100 [8] | N/A | Synthetic | Real |
| RealSR [5] | N/A | Real | Real |
| ImagePairs [11] | N/A | Real | Real |
| CUFED5 [27] | Real | Synthetic | Real |
| CameraFusion [20] | Real | Synthetic | Real |
| MiddlyBury [17] | Real | Synthetic | Real |
| Our ReWiTe | Real | Real | Real |

## 2.2 Related methods

Although the **W** and **T** dual-camera image fusion problem does not need to enhance the pixel resolution, most of existing methods are super-resolution methods.

The mainstream existing methods are supervised methods, including DCSR [20] and FaceDeblurSig [12], TTSR [24], SRNTT [26], MASA [16] and Shim20 [18]. They have to be trained on synthetic datasets currently due to the lack of real data for supervision. Additionally, they exhibit insufficient control over artifacts.

**Table 2: Camera parameters of OPPO Find X6 that are used in our hardware.**

| Camera | **T** of cellphone2 | **W** of cellphone2 | **T** of cellphone1 |
|---|---|---|---|
| Sensor | 1/1.56 inch | 1/1.56 inch | 1/1.56 inch |
| Pixel size | 1um | 1um | 1um |
| Resolution | 50mp | 50mp | 50mp |
| FOV | 36° | 84° | 36° |
| Focal length | 65mm | 24mm | 65mm |

**Table 3: Image resolution in the ReWiTe dataset.**

| | |
|---|---|
| **T** input | 3496*2472 |
| **W** input | 3496*2472 |
| **GT** | 3496*2472 |

Self-supervised learning, e.g. Selfedzsr [28] and Zedusr [23], address the issue of lacking real data in supervised learning. They propose warping **T** images to **W** perspectives as ground truth for network training. However, the inherent occlusion challenges in native **T** and **W** images remain unresolved. In addition, these methods often impose certain requirements on the model. Not all backbone models can be easily integrated into self-supervised networks.

Stereo super-resolution methods, like StereoSR [9] and PASSR [19], use homogeneous dual-camera configurations as input, deviating from the current mainstream camera setups. The enhancement potential of homogeneous configurations is less than that of heterogeneous configurations, and they also face the challenge of lacking real datasets.

In single-image super-resolution methods, benefiting from real datasets such as ImagePairs [11], models can have effective training. In self-supervised/unsupervised learning methods, GANs [21, 22] surpass the quality limits of ground truth theoretically but are prone to generating artifacts, conflicting with the requirements of mobile imaging applications. To solve the problem in this paper, single-image super-resolution methods fail to use the high-quality reference **T** images.

## 3 REWITE DATASET

The building of the ReWiTe dataset includes 1) the beam splitter camera rig based hardware design, and 2) the calibration for the built hardware and the shot images.

### 3.1 Hardware Design

The hardware design objective is to enable three cameras to simultaneously capture images and ensure that the shot input **W** image and the shot **GT** image share the same optical path. The primary challenge lies in constructing the beam splitter-based camera rig.

One possible approach for spectral division involves inserting a beamsplitter between the lens and the sensor [7]. However, this method cannot use commercial phone cameras as the camera of the dataset system because it requires a complete reconstruction of the entire camera components, including the lens, sensor, ISP, etc. The advantage of this approach is that the calibration of the spectral division system becomes somewhat simpler. However, the imaging effects produced by these reconstructed components are difficult to maintain consistent with the quality of real commercial cellphones. Because the final usage of dual camera fusion is mostly for phone camera systems, we did not adopt this approach.

Another approach to spectral division is to split the light before the lenses of commercially available cellphone cameras, which is the approach we have taken. This increases the difficulty of calibration as each of the three hardware components—the two cellphones and the beamsplitter—has six degrees of freedom in motion. They need to be jointly calibrated to the accurate spectral path. Despite the calibration challenges, this approach allows to obtain the most authentic images because the entire imaging process occurs within the commercially available cellphones. Apart from a halving of light intensity due to spectral division, the rest of the imaging process is identical to capturing images using a cellphone in real-world scenarios. Thus, we adopted this approach.

Our hardware consists of two cellphones, a beamsplitter, and a fixation device. The optical path of the three cameras on the two cellphones using our hardware is shown in Fig. 4. The 3D model of our hardware is shown in Fig. 5, and the real hardware is shown in Fig. 6. The important parts of our hardware are detailed below.

- Cellphones: We use two OPPO Find X6 cellphones. The camera parameters are shown in Table 2. The cellphone No. 2 serves for capturing the input **W** and **T** images simultaneously, named $W_2$ and $T_2$. The optical path of the **T** camera of cellphone No. 1 is aligned with the **W** camera of cellphone No. 2, and the image shot by this **T** camera, named $T_1$, is used to generate the ground truth image of the input **W** image. The parameters of the **W** and **T** cameras are shown in Table 2. The censors of the two cameras are the same, but the lens are different.
- The beamsplitter: It is a cube with the size of $60mm \times 60mm \times 60mm$. The beam splitter divides the optical path into a 50%-50% split, allowing the cameras on both sides of the beam splitter to capture images from the same optical path.
- Fixation device: The fixation device includes beamsplitter fixation and cellphones fixation. They ensure stability during shooting.
- Position adjustment knobs: The front-back, left-right, and up-down adjustment knobs can adjust the two cellphones in six directions, so as to enable us to align the **T** camera of cellphone No. 1 with the **W** camera of cellphone No. 2 during the calibration.

We utilize the **W** and **T** cameras of the OPPO Find X6 cellphones for our study, without considering other industrial cameras or DSLR cameras. This decision is driven by the widespread usage of smartphone cameras today. The potential application of **W** and **T** dual-camera image fusion is predominantly in smartphones, which already possess the necessary dual-camera systems, computational resources, and high demand from users. Different camera choices would result in varying quality differences between dual-camera images. To ensure that our dataset corresponds to practical scenarios, we opt to use phone cameras to capture the images and construct the dataset.

We have root access on the OPPO Find X6 cellphones that are used in our hardware, and we deliberately refrain from letting the cellphones run the enhancement methods inside the phone's imaging software to avoid them changing the image quality.

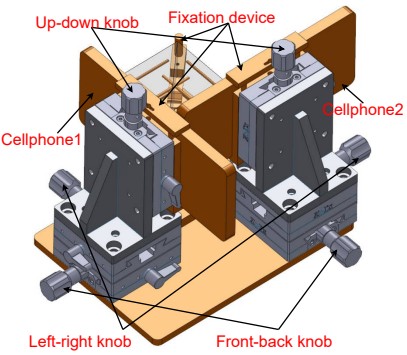

**Figure 4: Design diagram of our hardware and the spectral division system. It consists of two cellphones with T and W dual cameras, a beamsplitter to divide the optical path into a 50%-50% split, and the fixation device. We perform calibration to let the T camera of cellphone 1 share the same optical path of the W camera of cellphone 2. 'Wide' is short for the wide-angle camera, and 'Tele' is short for the telephoto camera.**

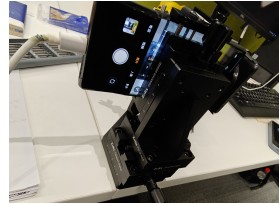
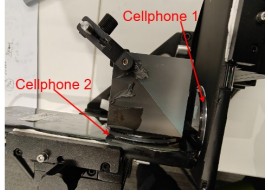

**Figure 5: 3D structure of our hardware. The six knobs are designed to perform the calibration.**

**Figure 6: The real photos of our hardware. We use two OPPO Find X6 cellphones to build the hardware.**

To enable simultaneous image capture by the **W** and **T** cameras on cellphone No. 2, we developed a synchronization control software on the cellphone. To enable simultaneous image capture by the cameras on cellphone No. 2 and the **T** camera on cellphone No. 1, we utilize Bluetooth shutters to control the shutters of the cellphones.

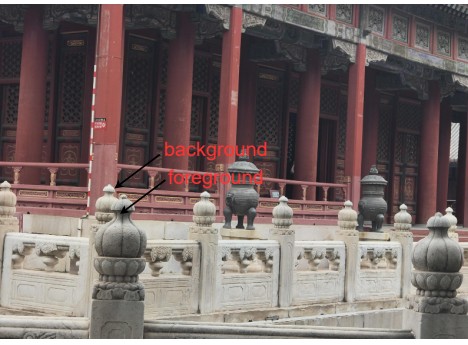

**Figure 7: An example of selected calibration scenes with rich occlusions, where we perform coarse and fine alignment to ensure that the shot images from the W camera of cellphone 2 and the T camera of cellphone 1 have no occlusions.**

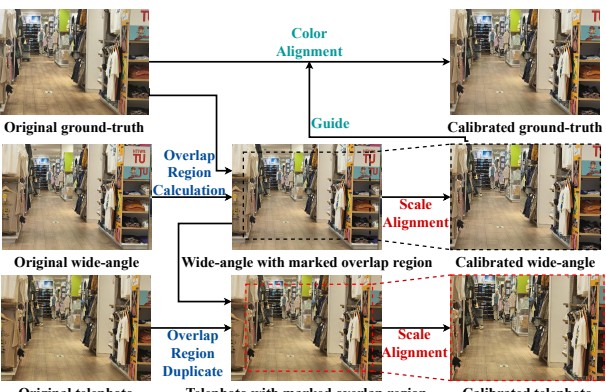

**Figure 8: The process of scale alignment and color alignment during the calibration. In the scale alignment, we calculate the overlap regions between the original Wide-angle (W) and GT images. The overlap regions of the original W image is upsampled to the size of the original GT image, so as to make the calibrated W image and the GT image have the same FOV. The overlap region is duplicated for the original Telephoto (T) image to perform its scale alignment, so as to keep the FOV differences of the W and T images not changed after the calibration. In the color alignment, to address tonal differences between the calibrated W image and the original GT image, using the calibrated W image as the guide, the original GT image is tone mapped to obtain the calibrated GT image.**

## 3.2 Calibration

After setting up the hardware, we proceed with calibration to generate images for the ReWiTe dataset. Firstly, we conduct coarse and fine hardware alignment to ensure that the W camera of cellphone 2 capturing the input **W** image and the **T** camera of cellphone 1 capturing the **GT** image share precisely the same optical path, and that there are no occlusions between the captured **W** and **GT** images. We select scenes with rich occlusions to perform the coarse and fine alignment, as Fig. 7 shows. Secondly, as illustrated in Fig. 8, we perform scale alignment and color alignment, to ensure that the calibrated **W** and **GT** images have the same FOV and color

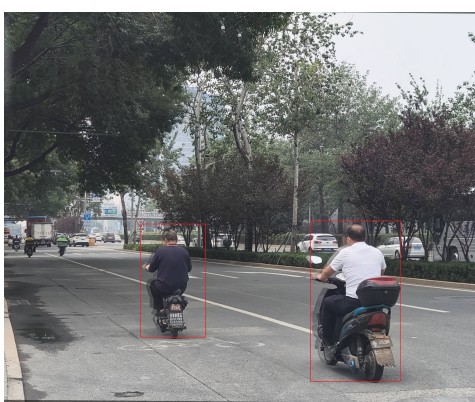

**Figure 9: An example of the marked problematic regions during the manual annotation.**

tone, while maintaining consistent FOV differences between the calibrated **W** and **T** images as observed in the original **W** and **T** images. Thirdly, as demonstrated in Fig. 9, we conduct manual annotation to mark problematic regions where the **GT** image provides incorrect annotation for the input **W** image.

- Coarse alignment: The fundamental purpose of the alignment is to make the input **W** image and the ground truth image without occlusion. We perform coarse alignment by using the knobs to move the position and shooting angle of the cellphone No. 1 and cellphone No. 2. As shown in Figure 7, we select scenes with rich occlusions to perform the alignment.
- Fine alignment: After the coarse alignment, we fix the cellphone No. 2 and continue to adjust the knobs of cellphone No. 1. By observing the occlusion information between $\mathbf{W}_2$ and $\mathbf{T}_1$ at different knob positions, we continuously adjust the position of cellphone No. 1, until the occlusions do not appear.
- Scale alignment: Through the operations in the above two steps, we have obtained occlusion-free $\mathbf{W}_2$ and $\mathbf{T}_1$. However, since they are captured by cameras with different FOVs, the image scales are different, and, in the un-overlapping regions, the $\mathbf{T}_1$ image cannot provide the corresponding **GT** information for the input **W** image. We perform scale alignment to make the sizes of the same objects in $\mathbf{W}_2$ and $\mathbf{T}_1$ consistent so that the **GT** image can provide the corresponding annotation for every pixel of the calibrated **W** image. We perform this alignment by calculating SIFT features [15] on both $\mathbf{W}_2$ and $\mathbf{T}_1$, finding overlap regions between similar features, and computing a projection transformation matrix to enlarge the overlap region of $\mathbf{W}_2$. To keep the FOV differences between the calibrated **W** and **T** images consistent with the FOV differences between the original **W** and **T** images, we duplicate the overlap region to the original **T** image and enlarge the overlap region of the original **T** image with the same transformation of the original **W** image.
- Color alignment: Due to differences of the capturing devices, $\mathbf{W}_2$ and $\mathbf{T}_1$ exhibit substantial color variations. We employed a 3D color tone adjustment method, calculating a $256 \times 256 \times$

256 color mapping matrix to adjust the color tone of $\mathbf{T}_1$.

$$\mathbf{T}_1^{aligned}(j, i) = \alpha_{\mathbf{T}_1, \mathbf{W}_2}(\mathbf{T}_1(j, i)) \tag{1}$$

$$\alpha_{\mathbf{T}_1, \mathbf{W}_2} = \frac{\sum_{(j,i) \in \omega(\mathbf{k})} \mathbf{W}_2(j, i)}{|\omega(\mathbf{k})|} \tag{2}$$

where $\omega(\mathbf{k})$ is the set of pixels in $\mathbf{T}_1$ whose intensities are equal to $\mathbf{k}$, and $|\omega(\mathbf{k})|$ is the number of pixels in the set $\omega(\mathbf{k})$.

- Image cropping: The images captured by cellphones have a resolution of 4K. However, the boundary regions of these images may exhibit darkening due to vignetting effects. To mitigate the impact of these effects on image quality, we crop out the boundary regions of the captured images, resulting in a cropped resolution of $3496 \times 2472$, as shown in Table 3.
- Manual annotation: As shown in the Figure 9, our image capture process may encounter situations where some regions are not calibrated correctly, such as fast-moving objects, calibration errors, or defocusing of **T** and **GT** images. To prevent these regions from impacting the quality of our training data, we conduct manual annotation to mark out these problematic areas.

In scale alignment, there are two approaches: upsampling the overlap region of the **W** image to match the size of the **GT** image or downsampling the **GT** image to match the size of the overlap region of the **W** image. Both methods accomplish scale alignment, but the downsampling approach sacrifices details in the **GT** image, resulting in smaller quality differences between the **W** input and **GT** images compared to the upsampling approach. To ensure that the training data reflects higher quality differences between the input **W** and **GT** images, we select the upsampling approach.

## 4 EXPERIMENTAL RESULTS

### 4.1 Comparison algorithms

The comparison methods we select are the SOTA dual camera image fusion/super-resolution methods, including DCSR [20], SelfDZSR [28], ZeDUSR [23], the SOTA reference-based super-resolution methods, including C2-matching [10], , MASA [16], TTSR [24], SRNTT [26], and the SOTA single-image super resolution methods, including RCAN [25], EDSR [14], Real-ESRGAN [22].

### 4.2 Experimental protocol

For dual-camera imaging in cellphones, the task primarily belongs to image fusion and enhancement. Both the input **W** and **T** images have the same resolution, such as 4K, and the desired output should also maintain this resolution, e.g. 4K. There is typically no requirement from users to upscale the resolution to 8K, 16K, or higher. Therefore, when constructing the ReWiTe dataset, we ensure that the input **W**, input **T**, and the **GT** images have identical resolutions, as indicated in Table 3.

However, in many comparison algorithms discussed in Section 4.1, the models default to increasing the pixel resolution of the input image, such as 2x or 4x upsampling, and the upsampling ratio remains fixed unless the model structure is modified.

For fair comparison, as depicted in Fig. 11, we adhere to these methods by downsampling the input **W** images by a factor of four to generate inputs for the comparison algorithms.

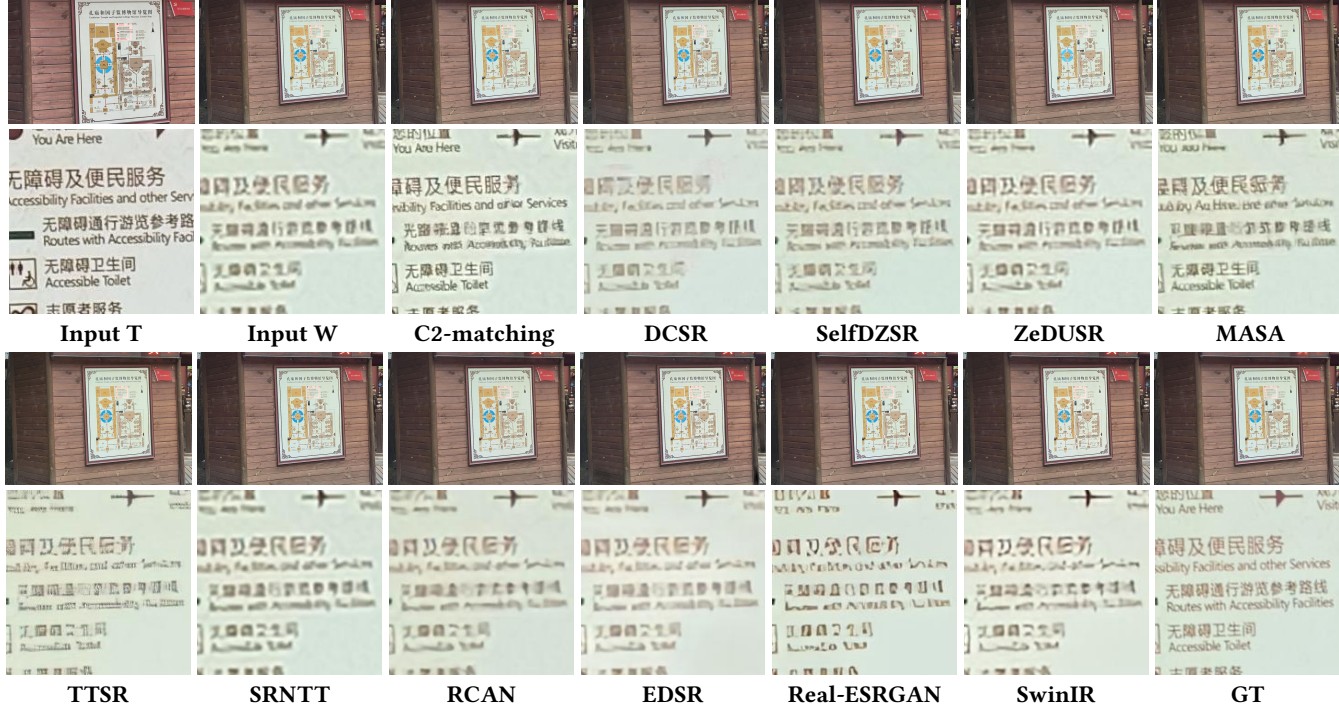

| Input T | Input W | C2-matching | DCSR | SelfDZSR | ZeDUSR | MASA |
| --- | --- | --- | --- | --- | --- | --- |

| TTSR | SRNTT | RCAN | EDSR | Real-ESRGAN | SwinIR | GT |
| --- | --- | --- | --- | --- | --- | --- |

Figure 10: Example results of different algorithms trained on ReWiTe. Top: The full-image results. Bottom: Results focused on the red-box region. Please see more results in supplementary materials.

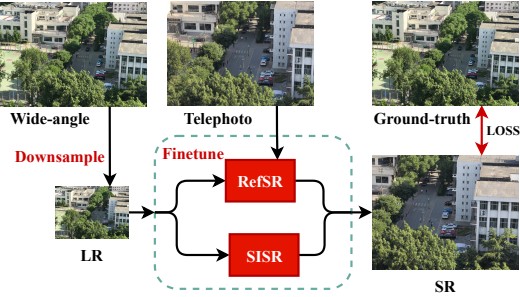

Figure 11: The experimental process to test the comparison algorithms. Because many comparison algorithms perform pixel resolution enlargement by default, for fair comparison, we follow their experimental pipeline. We downsample the input Wide-angel (W) image of ReWiTe with 4 times and use the downsampled images as the input of the algorithm. We use the input Telephoto (T) image of ReWiTe as the reference, and the Ground-truth (GT) image of ReWiTe as the GT.

## 4.3 Results

Table 4 and Fig. 10 show the quantitative and qualitative results of different comparison algorithms. We also show the example results of algorithms trained on CameraFusion vs. ReWiTe in Fig. 12.

As shown, when the training data transfers from the synthetic dataset of CameraFusion to our realistic dataset of ReWiTe, all the comparison algorithms obtain positive quality improvements. This verifies the benefits of the proposed ReWiTe dataset for various algorithms in fusing real **W** and **T** images.

Table 4: Average PSNR (dB)/SSIM values of different methods on ReWiTe. CF is short for CameraFusion. These methods are trained on the training set of CameraFusion and ReWiTe, respectively. They are tested on the testing set of ReWiTe.

| Methods | PSNR | SSIM | PSNR | SSIM |
| --- | --- | --- | --- | --- |
| | ReWiTe (Trained on CF) | | ReWiTe (Trained on ReWiTe) | |
| C2-matching [10] | 23.9730 | 0.8142 | 24.1500 | 0.8222 |
| DCSR [20] | 24.7196 | 0.8134 | **25.9661** | 0.8445 |
| SelfDZSR [28] | **24.9197** | **0.8385** | 25.9197 | **0.8510** |
| ZeDUSR [23] | 24.1709 | 0.8135 | 25.2618 | 0.8369 |
| MASA [16] | 23.7011 | 0.7977 | 24.9713 | 0.8154 |
| TTSR [24] | 22.1456 | 0.7188 | 25.5980 | 0.8286 |
| SRNTT [26] | 23.5913 | 0.8105 | 24.3436 | 0.8124 |
| RCAN [25] | 24.1709 | 0.8138 | 25.1189 | 0.8371 |
| EDSR [14] | 24.3540 | 0.8189 | 25.4792 | 0.8355 |
| Real-ESRGAN [22] | 22.1188 | 0.7465 | 24.4143 | 0.8192 |
| SwinIR [13] | 22.4462 | 0.7848 | 24.7440 | 0.8277 |

Among the results, the performances of dual camera image fusion/super-resolution methods, i.e. DCSR, SelfDZSR, and ZeDUSR, unsurprisingly achieve the best results. They utilize both the input **W** image and the input **T** image to estimate the output and are typically designed to solve this task.

Single-image super-resolution algorithms, i.e. RCAN, EDSR, Real-ESRGAN, and SwinIR, are not competitive to dual camera image fusion/super-resolution methods. This is because these algorithms only utilize the input **W** image to super-resolve the output without leveraging the input **T** image at all. This observation underscores the advantages of incorporating the input **T** image in addressing this problem.

Reference-based super-resolution algorithms, i.e. C2-matching, MASA, TTSR, and SRNTT, utilize both input **W** images and input

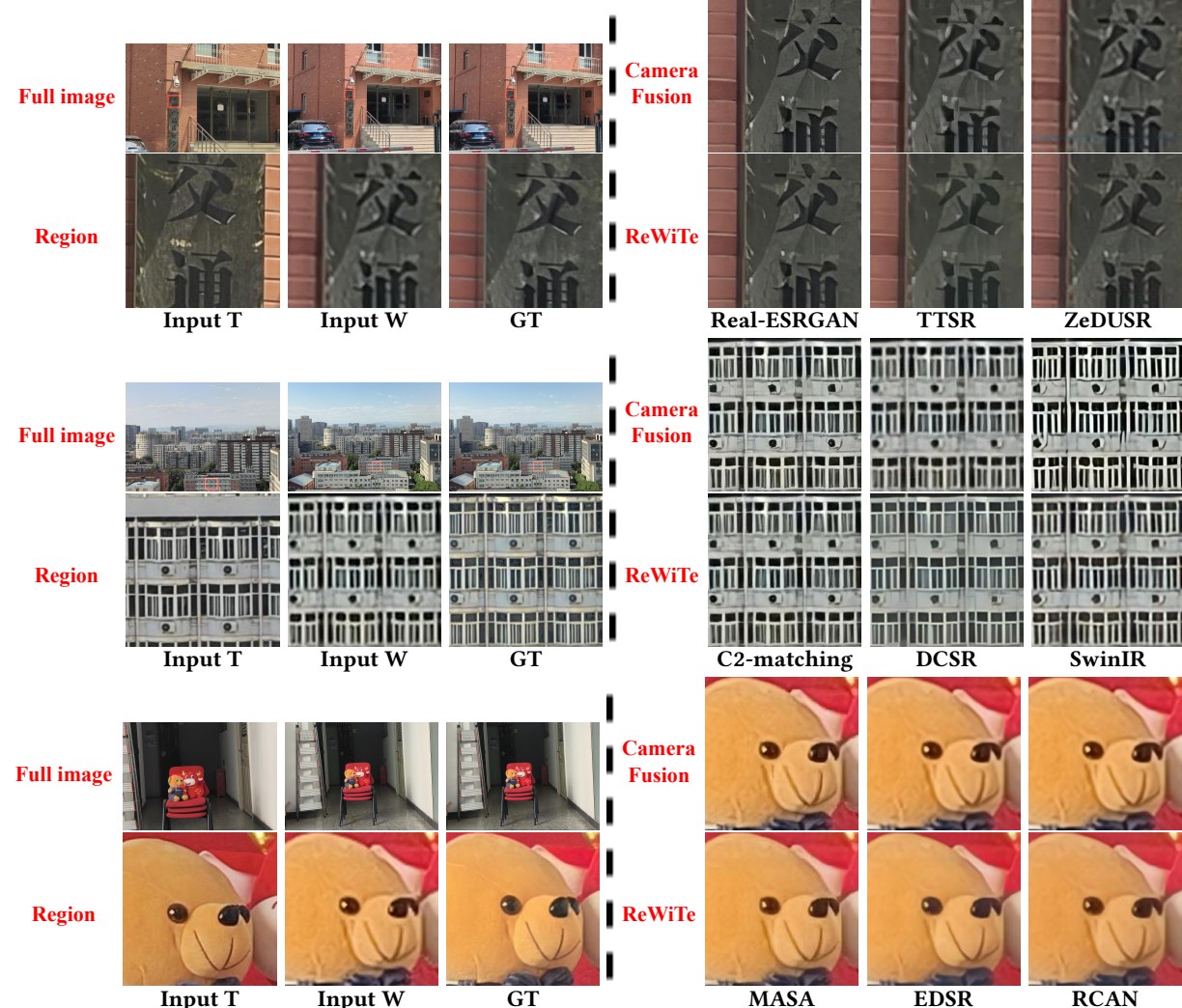

**Figure 12: Example results of different algorithms trained on CameraFusion vs. ReWiTe. In the left part, we show the full image and the red-box region of the input T and W images and the GT images. In the right part, we show the results of different algorithms on the red-box region, derived from the algorithms trained on CameraFusion vs. ReWiTe.**

**T** images to estimate the output. However, the their performances fall below dual-camera image fusion algorithms, and are even lower than some single-image super-resolution algorithms. The reason is that, due to the assumptions of input and reference images shot in different scenes, positions, and time, they usually try to transfer more high-quality details from the reference image into the input low-resolution image, even if the textures of the two images are not exactly the same. While this can usually improve the visual quality, the structure change of textures or even artifacts may be introduced into the results, leading to loss of PSNR/SSIM values.

## 5 CONCLUSIONS

This paper introduces a realistic **W** and **T** dual camera fusion dataset, named ReWiTe. It is created using the **W** and **T** cameras of one cellphone to capture the input pair of **W** and **T** images, while the **T** camera of another cellphone is used to capture the **GT**

image. A hardware setup employing a beam-splitter is designed, and a series of calibration processes are conducted to ensure that the input **W** image and the **GT** image share the same optical path. Consequently, the **GT** image maintains consistent quality with the input **T** image and provides annotations for the input **W** image at every pixel. Experimental results demonstrate the effectiveness of the ReWiTe dataset in enhancing various existing methods for the real-world **W** and **T** dual camera fusion task.

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
