# OpenReview forum: "ReWiTe: Realistic Wide-angle and Telephoto Dual Camera Fusion Dataset via Beam Splitter Camera Rig"
_acmmm.org/ACMMM/2024/Conference — MM2024 Poster_

### Official Review · Reviewer_e7pD · 2024-05-16

**Rating:** 3
**Confidence:** 2

**Summary:**

This paper introduces the ReWiTe dataset, a new method for fusing wide-angle and telephoto images using a novel hardware setup involving a beam splitter and dual cellphones. Unlike existing approaches that rely on synthetic datasets, ReWiTe captures high-quality, realistic images, providing accurate pixel-level annotations. Experiments demonstrate that models trained on ReWiTe perform better in image fusion and super-resolution tasks, highlighting the dataset's potential to improve real-world applications.

**Strengths:**

1. The ReWiTe dataset is the first to provide realistic wide-angle and telephoto images, addressing the limitations of synthetic datasets and improving model training for dual-camera image fusion.

2. Using an interesting setup, including beam splitter and dual cellphones, the dataset captures telephoto and wide-angle images along with a high-quality ground-truth image, ensuring accurate pixel-level annotations and consistent image quality.

**Limitations:**

1. The need for precise calibration to ensure accurate alignment of images could introduce variability and errors, affecting the consistency and reliability of the dataset.
2. Although innovative, the dataset contains only 342 image sets, which might be insufficient for training deep learning models at scale.
3. The evaluation provided may not sufficiently emphasize the dataset's contribution, lacking comprehensive comparisons with a wide range of existing methods and datasets to solidify its impact.
4. It is unclear whether the dataset will be made open source.

**Suitability:**

3

---

### Official Review · Reviewer_X8Ui · 2024-05-23

**Rating:** 1
**Confidence:** 2

**Summary:**

This paper introduces a hardware setup utilizing a beam splitter to simultaneously capture three images. It captures wide-angle and telephoto images, as well as ground-truth images, from two authentic cell phones equipped with cameras. The contribution is the hardware setup.

**Strengths:**

The contribution is the hardware setup that can capture three images together. The hardware includes two cellphones, a beamsplitter,
and a fixation device. With the hardware, they construct a realistic dataset.

**Limitations:**

1. I cannot understand how to use the beamsplitter to get the same scene. From fig.4, it looks like two cameras are perpendicular to each other, then how to get the same contents?
2. It is more like an engineering work with some tricks.
3. In Table 4, when the training data transfers from the synthetic dataset of CameraFusion to our realistic dataset of ReWiTe, all the comparison algorithms obtain positive quality improvements. I think it is obvious that when training and testing are the same(ReWiTe), there will be gains. But, we cannot conclude that these gains are provided by the real data.

**Suitability:**

2

---

### Official Review · Reviewer_1BuQ · 2024-05-24

**Rating:** 5
**Confidence:** 4

**Summary:**

The fusion of combining images from wide FoV and telephoto lenses results in high quality images with wide FoV. This kind of fusion is especially beneficial of mobile devices. Although many deep learning based method utilize synthetic dataset to train the model, and the real word data is mostly kept internally within the companies developing the smartphones. To speed up the development in this field, the authors have devised a setup to capture the dataset to train these models using commercially available smartphone camera, and a highly precise beam splitter mounting station which coupled with the calibration gives accurate ground truth and image pairs.

**Strengths:**

**1.** Its the first realistic dataset designed specifically for the telephoto and wide-angle dual-camera image fusion task. This is significant in a field where most datasets are synthetic and may not adequately capture real-world variations. Thus, the dataset is novel.

**2.** The beam-splitter camera rig to ensure that the ground truth (GT) image shares the same optical path as the wide-angle (W) image is a sophisticated technique. This ensures high-quality, realistic GT data that is directly comparable to the input data, enhancing the reliability of the dataset for training and evaluation.

**3.** The detailed calibration process to align images and mitigate differences in color and scale demonstrates a thorough approach to dataset creation.

**4.** As shown in results in the paper as well as visual results in supplementary materials training on the ReWiTe dataset improves the performance of various state-of-the-art methods compared to synthetic datasets. This practical validation highlights the dataset's potential to advance the field significantly.

**5.** The dataset consists of 342 sets of images, providing a substantial amount of data for both training and testing purposes. This volume and variety enhance the dataset's utility and applicability.

**Limitations:**

**1. (Minor)** The creation of the dataset relies heavily on a specific hardware setup involving multiple cellphones and a beam-splitter. This complexity could limit the replicability of the dataset creation process (Unless the 3D structure is opensourced)

**2.** Although 342 image sets provide a good starting point (especially for fine tuning an already trained model), the dataset size might still be relatively small for training deep learning models that require large amounts of data. Increasing the dataset size could improve the robustness and generalizability of models trained on it. Although authors can iteratively add more images in future to increase the size of dataset.

**Suitability:**

3

---

### Meta-Review · Area_Chair_Bhq6 · 2024-07-05

**Recommendation:** Accept (Poster)
**Confidence:** 5

**Metareview:**

This paper received 1 Weak Accept, 1 Reject, 1 Borderline Reject in the first round of review. After rebuttal, one reviewer remained unchanged (Weak Accept), and two reviewer slightly raised the scores from Reject to Borderline Reject, and Borderline Reject to Borderline Accept. The concerns remaining are on the realistic value of the dataset, and the appropriateness of the experiment comparsion. Given the situation of mixed scores, AC carefully examined all materials at hand. In addition to the issues pointed out by reviewers, AC also noticed the minor color discrepancy between input W and the created GT. By discussing this paper with SAC, AC made a hard decision to accept it. Yet, the authors are required to improve the manuscript substantially, as they said in the rebuttal.